# Research on trust mechanism of supply chain finance under Industrial Internet embedded with blockchain

Yingmei Jiang[1], Yuxin Li[2], Jinyu Wei[2], Yaoxi Liu 🔟 [2]*

1 Business School, Jinhua Polytechnic, Jinhua, Zhejiang, China, 2 School of Management, Tianjin University of Technology, Tianjin, China

* liu405883443@163.com

## Abstract

In traditional supply chain finance, the financing of enterprise mainly relies on the credit segmentation of the core enterprise, resulting in a short trust transmission radius and poor financing ability. The development of Internet technology, while expanding financing channels, has also seen an increasing severity in issues such as information fraud and data breaches, which has further aggravated the trust crisis in supply chain finance. This paper integrates blockchain technology into the industrial internet platform and analyzes the applicability of both in empowering supply chain financial trust. Then a supply chain financial trust framework, which emphasizes information sharing, data security, and trust circulation, is proposed. Furthermore, combined with the theories of Funk-SVD and entropy value, this paper designs a global trust evaluation mechanism that facilitates the trust circulation in supply chain finance and proposes a recommendation algorithm for global trust. With the testing conducted using the Epinions dataset, it is found that the algorithm proposed in this paper has a strong data dimensionality reduction and concentration ability, especially for large sample data, it can obtain more accurate evaluation values with less space occupation, thus enhancing the trust circulation ability of supply chain finance. Finally, the paper puts forward specific policy recommendations for the implementation of the supply chain finance information mechanism, aiming to better improve the financing accessibility of enterprises in supply chain, particularly small and medium-sized enterprises.

## 1 Introduction

For any country, small and medium-sized enterprises (SMEs) are a crucial force that cannot be ignored in its economic development process. In order to support the development of SMEs, countries around the world have also taken a series of measures to improve the accessibility of financing for SMEs. For example, the US government provides financial loans to SMEs at lower interest rates than the market loans through the establishment of fiscal loans, in order to alleviate the burden on enterprises. The German government, as the leading institution, has collaborated with commercial banks, guarantee companies, and policy banks to establish a systematic financing system and regulatory model for SMEs, in order to balance the conflicting

**Data Availability Statement:** All relevant data are within the manuscript and its Supporting Information files.

**Funding:** This paper is supported by the 2022 Humanities and Social Science Youth Fund of the

Ministry of Education (Grant No. 22YJC630046). The recipient is Yingmei Jiang. The main role of the funder Yingmei Jiang in this manuscript is as follows: study design, funding acquisition, data analysis, and Writing – review & editing.

**Competing interests:** The authors have declared that no competing interests exist.

interests of various parties in the financing process. The Chinese government has also introduced a series of fiscal measures to guide and establish new financing models for SMEs. As a result, supply chain finance, which aims to support the development of SMEs, has gradually become a topic of public interest and is constantly evolving as a new direction in supply chain theory.

Supply chain finance is a financing model that relies on the credit of the core enterprise and is formed through the collaboration of multiple parties, including the core enterprise, upstream and downstream SMEs, and financial institutions, in order to create a universal, recyclable, and efficient financing model. Compared with traditional financing models, supply chain finance innovatively introduces the transmission of core enterprise credit, which helps to enhance the credit of SMEs. This enables them to obtain more favorable interest rates and shorter financing time, to a certain extent alleviating the financing constraints faced by SMEs.

However, in the current development process of supply chain finance, the credit risk of SMEs is considered to be one of the most important and severe risks. On the one hand, SMEs often lack core technology and innovation capabilities, resulting in smaller scale, weaker strength, and limited asset quality. This makes it difficult for them to provide sufficient collateral, leading financial institutions to adopt strict financing conditions to control bad debt risks. On the other hand, supply chain companies often hide certain true information or even disseminate incorrect information in their daily transactions due to different purposes. This asymmetric information not only affects the overall operational efficiency of the supply chain but also increases the risk for financial institutions, leading to a crisis of trust in corporate financing. In fact, SMEs still face difficulties in obtaining financing and high financing costs.

In addition, the rapid development of industrial Internet technology not only provides a massive data source for the comprehensive evaluation of corporate credibility, but also exposes the deficiencies of financing platforms in terms of data processing capabilities. The inability to extract effective information for credit assessment of SMEs from massive data, as well as the lack of accurate and timely communication, significantly hinders the realization of financing for SMEs and ultimately affects their overall development. Hence, in the context of industrial interconnection, accurately transmitting trust data in the supply chain and designing a scientifically rational mechanism for evaluating trust in SMEs is not only crucial for solving the financing problem of these enterprises, but also for promoting the development of the supply chain and the supply chain finance.

Taking the above considerations into account, this article conducts research on the trust issues in supply chain finance by embedding blockchain technology into the Industrial Internet platform. It primarily discusses three questions: What is the architecture of supply chain finance under the integration of blockchain technology and the Industrial Internet platform? What is the trust evaluation mechanism for supply chain finance under this architecture, and how effective is it? How can we promote the implementation of the trust mechanism for supply chain finance by integrating blockchain and the Industrial Internet?

Based on the above issues, this paper comprehensively explores the applicability and role of Industrial Internet technology and blockchain technology in supporting supply chain finance activities, and constructs a supply chain finance platform architecture that integrates blockchain technology and industrial internet. By utilizing the distributed data storage, dynamic transaction monitoring, and massive data modeling and analysis technologies of the Industrial Internet platform and blockchain, a global trust evaluation model that integrates Funk-SVD and information entropy is designed. Flexible and intelligent machine learning recommendation algorithm is used to generate a trust value that includes comprehensive trust information about financing companies. This aims to improve the accuracy of trust assessment for SMEs in supply chain, and alleviate the information asymmetry in supply chain finance. Finally, the

proposed algorithm is subjected to performance tests using a publicly available dataset, confirming its superiority in processing large sample data. Based on the analysis of the test results, corresponding countermeasures are proposed.

The main contributions and significance of this study are as follows: Firstly, by integrating blockchain technology with industrial Internet technology, the hierarchical architecture of supply chain financial trust is reconstructed, and the operational logic of the supply chain financial trust mechanism is proposed. Secondly, the recommendation algorithm based on Funk-SVD and information entropy is applied to the evaluation of supply chain financial trust under the industrial Internet, achieving a comprehensive trust evaluation of supply chain trust mechanism. Thirdly, this study proposes strategies and suggestions for improving the supply chain financial trust mechanism under the integration of industrial Internet and blockchain, which can help the supply chain improve the accessibility of financing for SMEs and effectively control the bad debt risk of financial institutions. In general, this research not only enriches the theoretical research of the supply chain and supply chain finance under the integration of the industrial Internet and blockchain, but also provides guidance for the financing practices of SMEs in the supply chain in the Internet era.

## 2 Literature review

### 2.1 Research on trust mechanism of supply chain finance

Timme first introduced the concept of supply chain finance in 1993, describing it as a collaboration between supply chain enterprises and external institutions aimed at providing financial support for supply chain business transactions [1]. This financial model can enhance the value of the supply chain and promote the flow of financial resources within it [2]. In 2009, Pfohl and Gomm [3] emphasized that supply chain finance assists in the capital cycle among supply chain enterprises and, compared to other commercial bank loans, can more effectively increase the profits of supply chain participants. Therefore, scholars generally regard supply chain finance as an essential service for promoting efficient capital flow in the supply chain [4]. Overall, introducing third-party financial institutions to manage the capital flow of the supply chain can minimize its financial costs [5].

However, as research on supply chain finance deepened, a problem gradually emerged: supply chain participants might conceal information in transactions to protect their competitive position. This information asymmetry can increase the risk to financial institutions and lead to trust issues. To address this challenge [6], Song(2017) [7] proposed that the trust issue in supply chain finance mainly arises from the information asymmetry between banks and enterprises. They suggest establishing a corporate credit assessment system and a credit default punishment mechanism and believe that the introduction of a third-party authoritative institution for supervision would be beneficial. Moretto A (2019) [8] believes that most of the current trust evaluation models mainly rely on financial data, which is difficult to achieve enterprise credit enhancement. Yao(2022) [9] constructed SVME-AIR and FS-MRI models considering the characteristics of enterprise categories to correct the trust data bias.

Currently, research on trust in supply chain finance is primarily focused on building credit risk models, and there has not yet been an in-depth study of the integration of various information technologies and supply chain finance trust mechanisms. In fact, a complete trust system for better dealing with information asymmetry in financing has not yet been established.

### 2.2 Research on the application of industrial Internet to supply chain finance

The concept of the Industrial Internet was first introduced by General Electric (GE) in 2012. It aims to utilize the latest internet technologies to establish an intelligent manufacturing and

interconnection system [10], thereby promoting the next industrial revolution and the rapid development of the digital economy. The potential of the Industrial Internet in financial empowerment is evident. Using advanced identification technology, the platform can track the progress and transaction status of supply chain business in real-time, providing robust conditions for risk warnings [11]. Through intelligent algorithms and big data technology, the Industrial Internet can quickly collect and process key data from supply chain enterprises, thereby enhancing information availability, shortening acquisition time, alleviating manual review pressures, and assisting financial institutions in obtaining reliable information from SMEs [12]. Moreover, the application layer of the Industrial Internet offers convenient business access to each node of the supply chain. It not only provides personalized order production services but also offers investment risk analysis for financial institutions, controls bad debt losses, and tracks subsequent loan repayments in supply chain finance, thus building an intelligent, digital supply chain platform [13]. In summary, the Industrial Internet shifts the basis of trust in financial services from between institutions and enterprises to machine trust. It ensures the interests of supply chain participants through data mining, modeling analysis, and information security technology, creating a healthy financial environment.

However, the efficient application of the Industrial Internet in manufacturing supply chains is still in the exploratory stage. Most of the research has focused mainly on intelligent manufacturing models and the transformation and upgrading of supply chain technologies, and there has been relatively less discussion on supply chain financial services, especially on financing and credit. In addition, some researches on the application of Industrial Internet in supply chain finance have focused more on the technical aspect, without yet exploring quantitative methods for supply chain finance trust.

## 2.3 Research on the integration of supply chain finance and blockchain technology

Blockchain technology entered public view with the advent of Bitcoin. This technology, leveraging timestamps and based on a consensus mechanism, can record immutable data in blocks and enable validation and sharing of all nodes on a decentralized platform [14]. Scholars have deeply researched the integration of supply chain finance and blockchain technology, defining the conceptual framework of the Blockchain-driven Supply Chain Financial Platform (BcSCFP) [15]. This framework not only offers an operational model for supply chain finance but also emphasizes the pivotal role of blockchain in ensuring data security and circulation of digital assets [16].

With deeper research, experts started focusing on the specific application and optimization of blockchain technology in supply chain finance. Du and Chen (2020) [17, 18] designed a supply chain model integrated into blockchain, using homomorphic encryption and distributed ledger technology to improve the supply chain trust environment; a Rijanto (2021) [19] applied blockchain identification traceability technology to solve the problems of fraud in accounts receivable and purchase orders;. liu(2021) [20] also explored the problem of financial information asymmetry in supply chain, and constructed a double-chain that is convenient for verifying enterprise information and data; Meanwhile, Guo (2022) [21] further combined the Internet of Things technology to build a BC4Regu supply chain information management framework. Qiu et al. [22] found that through distributed ledgers, data traceability, and smart contracts, blockchain significantly propelled the development of supply chain finance.

Many scholars also suggested improvements to the blockchain technology itself. Erik [23] believed that this technology could enhance supply chain transparency and reduce credit risk.

Chen et al. [24] posited that a blockchain-based credit mechanism could provide solutions to credit risks for SMEs in supply chain. Lin et al. [25] indicated that the features of blockchain, such as decentralization and the immutability of smart contracts, can reduce credit risks in supply chain finance.

Although the unique features of blockchain technology can fundamentally ensure the fairness and transparency of transactions and reduce credit risks in supply chain finance, researches on its integration with Industrial Internet are relatively limited, and there is no discussion on the construction of a trust framework for supply chain finance under their integration, as well as the analysis of the pathway.

## 3 The appropriateness analysis of blockchain-embedded Industrial Internet to empower supply chain financial trust

### 3.1 Multi-node cross-validation, identity and information cannot be forged

The authenticity of information on the chain is extremely crucial in supply chain finance. The blockchain-embedded industrial internet uses RSA asymmetric encryption, where each node can verify identity information with each other, preventing malicious nodes from accessing the system for illicit profit. Digital signature technology also ensures authenticity and security of supply chain finance participants, where transaction information between nodes is encrypted and broadcast by the community, and transaction nodes can verify the transaction information. Once the financing enterprise fabricates a fake transaction to obtain financing, it will be discovered by othernodes [22]. The asymmetric encryption technology of the Industrial Internet embedded with blockchain has laid the foundation for information trust in supply chain finance.

### 3.2 Distributed data ledger, data is secure and cannot be edited

Different from the existing supply chain financial platform model, the Industrial Internet platform embedded with blockchain uses distributed data processing technologies such as Hadoop, Spark, Storm, etc. The supply chain enterprise nodes cannot modify the database information, there will be no data loss or system paralysis due to the central server being attacked. In addition, the timestamp of the storage block makes it traceable when each piece of data is generated and updated, so that companies cannot fake data. Distributed ledgers ensure data security and reliability, which not only addresses the information island problem, but also alleviates the problem of highly centralized platforms.

### 3.3 Mass transaction data storage, instant information sharing and monitoring

Industrial Internet provides a low-latency and high-stability data acquisition and transmission network channel for equipment connectivity in supply chain enterprises. Through automatic information collection, avoid enterprise information fraud and human error. Secondly, the identification analysis system can enable asset tagging, product tracing and transaction tracking of supply chain enterprises, ensuring authenticity of transactions and clear ownership of mortgage assets; In addition, IaaS, PaaS and SaaS provide application portals for data storage and processing, cloud computing analytics, and real-time monitoring of business information and trust risk for financing enterprises. As shown in Fig 1, the Industrial Internet technology embedded with blockchain can provide layer-by-layer information sharing and data security guarantee for supply chain finance.

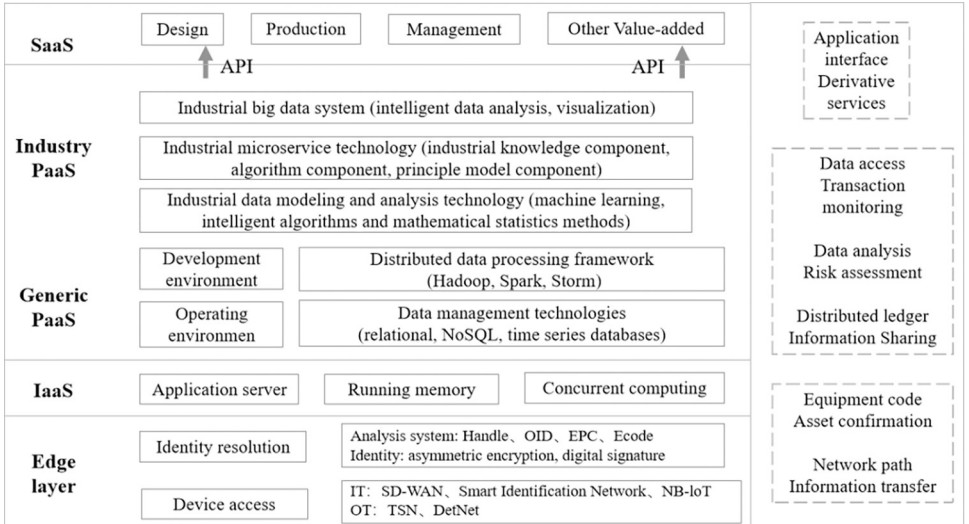

**Fig 1. The technical hierarchy structure of SFC platform under Industrial Internet.**

## 3.4 Self-execution of financing smart contracts, low cost and high efficiency of loanrepayment

The Industrial Internet supply chain finance platform architecture embedded with blockchain can intelligently control all elements and links of the supply chain through device perception, system ubiquitous connection, and data integration. Among them, the platform can use smart contracts to enable financing matching and loan repayment tracking for supply chain finance. In the case of loan repayment tracking in Fig 2, default is an essential factor in establishing trust between banks and enterprises, and the smart contract is controlled by the code. Once deployed, it will enforce, automatically judge the loan repayment situation of enterprises and guarantee loan recovery. In addition, smart contracts transform financial institutions' trust in businesses into machine trust, reducing the cost of manual audits and improving financing efficiency, enabling second-tier arrival of loans and debts, and creating powerful conditions for trust flow in supply chain finance.

To sum up, from the aspects of technology foundation, data security, information sharing and trust circulation, it is not difficult to find that the integration of the Industrial Internet of

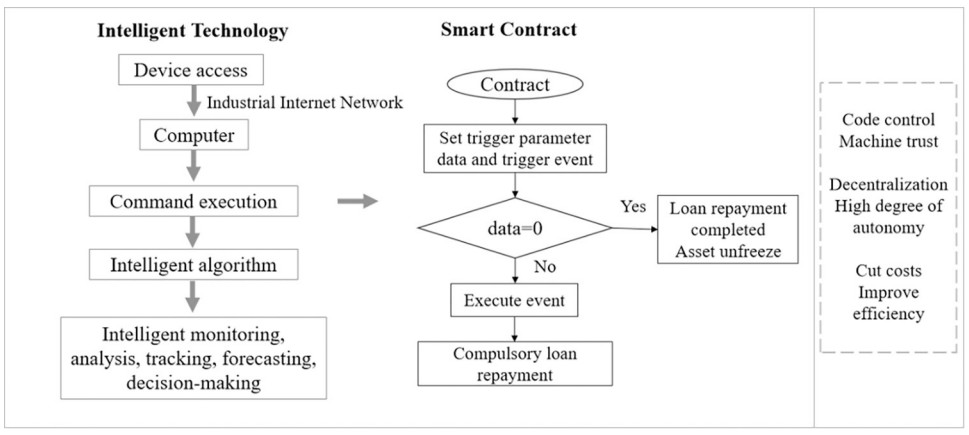

**Fig 2. Supply chain finance smart contract.**

the Internet of Everything and the "trust machine" blockchain technology can create a good information environment and trust environment for supply chain finance, and give full play to the value of supply chain data.

## 4 Design of supply chain finance trust evaluation mechanism based on Industrial Internet platform

The trust evaluation mechanism for supply chain finance in the industrial Internet embedded with blockchain mainly involves three parts: a evaluation principle, a trust consensus rule, and a evaluation algorithm. The trust evaluation principle is the way to obtain trust values of supply chain finance nodes and is the internal logic of the trust evaluation mechanism. The trust consensus rule ensures that all nodes of supply chain finance strictly adhere to the evaluation method. And it is also the premise for the effective circulation of trust values. The design of the evaluation algorithm directly determines the level of the calculated final trust value and embodies the rationality and scientificity of the evaluation mechanism.

Based on traditional trust evaluation theory, the trust evaluation principle and the consensus rule that are suitable for supply chain finance trust operation logic and blockchain technology features are designed. Furthermore, a global trust evaluation algorithm based on recommendation algorithms in the field of computer science and information entropy theory is proposed. The trust token calculated in this way can serve as a credit credential for financing enterprises to obtain financing from investment institutions, aiding SMEs in the supply chain to enhance their creditworthiness and facilitate trust circulation.

### 4.1 The trust evaluation principles of supply chain finance

The unified principle of trust evaluation is an important prerequisite for standardizing the behavior of trust evaluation. In order to obtain the trust value recognized by all nodes of the supply chain, it is necessary to consider the difference in the discourse power of each node and the timeliness of trust evaluation, and define the corresponding judgment criteria. The trust evaluation principle is as follows:

1. All supply chain nodes participate in the trust evaluation of each node.

2. The node newly added to the supply chain has no trust value until it interacts with the node on the chain and the counterparty makes an evaluation.

3. The nodes that have transaction records with the evaluated nodes in the past year shall be directly evaluated after the end of the last transaction, and the new evaluation value shall cover the old evaluation value. For the node that has no transaction record with the evaluated node in the past year, the trust evaluation value fails and returns to zero, and the indirect evaluation value is obtained through the recommendation algorithm with the evaluation value of other nodes.

4. The criterion for judging whether there is a transaction is that there are logistics, capital flow and business flow between nodes. Whether the final transaction is successful or not, the transaction is deemed to have occurred.

5. The criteria for judging the end of the transaction is that all the commodity and service activities of a certain contract have ended.

6. With the help of the information entropy model, the trust value weights of direct nodes and indirect nodes are allocated, the value after integrating the trust evaluation value by all nodes in supply chain finance is the trust token of the evaluated node.

## 4.2 Supply chain finance trust consensus rule

The trust consensus rule must be approved by all nodes. All nodes reach an agreement on the trust evaluation principle, trust value and trust supervision. The specific trust consensus rule is as follows:

1. All nodes agree on the principle of trust evaluation.

2. All nodes participate in the trust evaluation of supply chain finance and recognize the trust value of each node.

3. If 1/3 nodes do not recognize the trust value of a node, the re-evaluation procedure will be triggered. After a new round of evaluation, the re-voting process will be started until more than 2/3 nodes recognize the trust value of the node.

4. All nodes have the responsibility to supervise and maintain the accuracy of platform information, data security and trust circulation. If the node violates the system, once confirmed, the punishment will be implemented.

5. All nodes recognize the above rules.

## 4.3 Global trust evaluation algorithm based on Funk-SVD and information entropy model

The trust evaluation in supply chain can be divided into direct evaluation and indirect evaluation based on whether direct transactions occur. Direct evaluation occurs when there are direct transactions between nodes, and both supply and demand parties can evaluate each other based on their transaction experience. If there are no direct transactions between nodes, indirect evaluation is needed based on the recommendation algorithm, which is designed by Funk-SVD. Then combining with the information entropy model, a supply chain finance global trust evaluation algorithm suitable for integrating blockchain and industrial internet is proposed, thereby obtaining a global trust value that includes all trust information of supply chain enterprises. It should be noted that the direct trust evaluation value can be directly obtained by the service entrance of the Industrial Internet platform embedded with blockchain. The following mainly introduces the calculation method of the indirect trust evaluation value.

**4.3.1. Recommendation algorithm.** The principle of applying the recommendation algorithm to the trust evaluation of supply chain finance is as follows: by calculating the similarity of the common evaluation items between the indirect evaluation node and the direct evaluation node, can obtain the indirect trust evaluation value of the indirect evaluation node to the evaluated node. In this process, $I$ is the evaluated node, $u$ is the indirect evaluation node, $S$ is the set of direct trading nodes of $i$, $D$ is the set of nodes that directly trade with $U$, and $H$ is the set of nodes in $S$ that directly trade with any element in $D$, The trust evaluation matrix of $U$ to $D$ is $D_{1 \times n}$, and the trust evaluation matrix of $H$ to $D$ is $R_{m \times n}$. The similarity $\rho$ between $D_{1 \times n}$ and the trust vector of each node in $H$ to $D$ is obtained by calculation. Since $H$ has a direct trust evaluation value $V_z$ for $i$, the indirect trust evaluation value $v_{ui}$ of $U$ for $i$ can be approximated by combining $\rho$ and $V_z$.

**4.3.2. Funk-SVD.** In practice, the transaction range of many nodes on the supply chain is limited, and there are a large number of null values in the evaluation matrix. In order to obtain more accurate node evaluation similarity, data noise reduction and focus of sparse matrix should be carried out first. In this paper, the Funk-SVD method, which is improved based on SVD in the recommendation algorithm, is adopted. The assumption is that: the trust of the

evaluation node to the evaluated node is affected by the implicit classification of the evaluated node, each evaluated node may belong to many hidden classes, and has a specific degree of coincidence with the characteristics of each hidden class. In short, the more the evaluated node conforms to the trust category of the evaluated node, the more likely the evaluated node is to trust the node. The Funk-SVD expression is:

$$R^{\smile}_{m \times n} = P_{m \times k} Q_{n \times k}{}^{T} \tag{1}$$

Where, $k$ represents that the evaluated node has $k$ hidden class features, $P$ is the preference matrix of each hidden class of the evaluation node, and $Q$ is the coincidence matrix of the characteristics of the evaluated node and each hidden class. $R_{m \times n}$ is the sparse original trust evaluation matrix, $R^{\smile}_{m \times n}$ is the dense evaluation matrix after Funk-SVD processing. In order to minimize the data error, Funk-SVD uses the loss function to calculate the optimal solution:

$$L = \sum_{(h,i) \epsilon S} (r_{hi} - \tilde{r_{hi}})^2 = \sum_{(h,i) \epsilon S} \left( r_{hi} - \sum_{k=1}^{k} p_{hk} q_{ik} \right)^2 + \lambda \|p_{hk}\|^2 + \lambda \|q_{ik}\|^2 \tag{2}$$

Taking the loss function as the objective function and the hidden class factor $p_{hk}$ and the feature coincidence factor $q_{ik}$- as the independent variables, takes the partial derivative with respect to $p_{hk}$ and $q_{ik}$, set the initial values $\lambda$ and $\alpha$. The gradient descent method is used to search for the evaluation value that minimizes the error in the convergence direction of the loss function. In the iteration process, each element in matrix $P$ and $Q$ are updated continuously. When the error is less than the set accuracy, the iteration is stopped and the processed dense trust evaluation matrix is obtained.

$$l_{hi} = (r_{hi} - p_{hk} q_{ik})^2 \tag{3}$$

$$\frac{\partial l}{\partial p_{hk}} = -2 \left( r_{hi} - \sum_{k=1}^{k} p_{hk} q_{ik} \right) q_{ik} + 2\lambda p_{hk} = -2l_{hi} q_{ik} + 2\lambda p_{hk} \tag{4}$$

$$\frac{\partial l}{\partial q_{ik}} = -2 \left( r_{hi} - \sum_{k=1}^{k} p_{hk} q_{ik} \right) p_{hk} + 2\lambda q_{ik} = -2l_{hi} p_{hk} + 2\lambda q_{ik} \tag{5}$$

$$p_{hk} = p_{hk} - \alpha \frac{\partial l}{\partial p_{hk}} = p_{hk} + 2\alpha(l_{hi} q_{ik} - \lambda p_{hk}) \tag{6}$$

$$q_{ik} = q_{ik} - \alpha \frac{\partial l}{\partial q_{ik}} = q_{ik} + 2\alpha(l_{hi} p_{hk} - \lambda q_{ik}) \tag{7}$$

After obtaining the trust evaluation matrix after data processing, the score similarity between the indirect evaluation node and each element node of $H$ is calculated. Pearson method is an improvement of Euclidean and cosine similarity calculation method. Firstly, the two groups of vectors are centralized to avoid missing data in a certain dimension of the vector, and then the cosine similarity principle is used to find the spatial angle of the centralized vectors.

$$\rho(X, Y) = \frac{\sum (X - \bar{X})(Y - \bar{Y})}{\sqrt{\sum (X - \bar{X})^2 \sum (Y - \bar{Y})^2}} \tag{8}$$

By calculating the similarity between the evaluation matrix vector $D_{1 \times n}$ of $u$ and the evaluation matrix vector $R^{\smile}_{1 \times n}$ of node $h_{\mathrm{m}}$ in node set $H$, and combining the direct trust evaluation

value of the evaluated node, the indirect trust evaluation value of $u$ to $i$ can be obtained by weighted average:

$$v_{ui} = \frac{\sum(\rho_{um} \times v_{mi})}{m} \tag{9}$$

**4.3.3. Information entropy.** In the supply chain trust, the more transaction nodes passing through from the indirect evaluation node to the evaluated node, the lower the accuracy of trust transmission, and the weight of its evaluation value in the total evaluation value should be reduced accordingly. Therefore, the information entropy model is introduced to adjust the loss of trust. After parameter design, both direct and indirect nodes are considered, and a new information entropy weight model is proposed:

$$\omega_{ui} = \frac{2}{\log_2(Dis + 3)} \tag{10}$$

$Dis$ represents the shortest distance from the evaluation node to the evaluated node, the $Dis$ of the direct evaluation node is 1. In this model, the weight of the indirect evaluation node is always smaller than that of the direct evaluation node, and the weight becomes smaller and smaller with the increase of $Dis$, which conforms to the information entropy theory. On this basis, the formulas of direct trust evaluation value and indirect trust evaluation value are adjusted accordingly:

$$v_s = \omega_{si} v_{si} = \frac{2 \times v_{si}}{\log_2(Dis_{si} + 3)} \tag{11}$$

$$v_u = \omega_{ui} v_{ui} == \frac{2 \times v_{ui}}{\log_2(Dis_{ui} + 3)} \tag{12}$$

**4.3.4. Global trust value.** There are direct evaluation value set $V_s = \{v_1, v_2, \ldots, v_j\}$ and indirect evaluation value set $V_u = \{v_1, v_2, \ldots, v_g\}$. Due to the large number of nodes on the supply chain finance platform architecture, the weighted average calculation method is adopted here, and the range of node trust value is set as [0,1] by setting the interval of platform trust evaluation value, $t$ is the total number of nodes in the supply chain:

$$t = j + g \tag{13}$$

$$V_i = \frac{\sum_{i=1}^{t}(v)}{t} \tag{14}$$

## 5 Empirical analysis

In order to verify the realizability and performance superiority of the algorithm in this paper, we simulated the processes of obtaining trust evaluation matrix, matrix decomposition, similarity calculation, shortest distance calculation, weight adjustment, and global trust value calculation in Python 3.6.1 environment under Windows system. In addition, the Epinions dataset [26] is used to evaluate the performance of the proposed algorithm in solving trust evaluation data.

### 5.1 Algorithm symbols

For the convenience of reference, Table 1 describes the parameters and symbols involved in the trust evaluation of supply chain finance.

**Table 1. Parameters and symbols of trust evaluation algorithm.**

| Parameter | Symbol |
|---|---|
| The evaluated node | $i$ |
| Direct evaluation node set | $S$ |
| Indirect evaluation node set | $U$ |
| Direct evaluation node set of indirect evaluation nodes | $D$ |
| The set of nodes in $S$ that have direct transactions with any element in $D$ | $H$ |
| Weight of node trust evaluation value after loss of trust transmission | $W$ |
| The similarity between nodes in $U$ and nodes in $H$ | $\rho$ |
| The direct evaluation value of the evaluated node | $v_s$ |
| The indirect evaluation value of the evaluated node | $v_u$ |
| Trust value of the evaluated node | $v_i$ |

## 5.2 Algorithm steps

In order to obtain the global trust value of the evaluated node, after accessing the platform database to obtain the trust data, obtains the trust evaluation matrix, and calculates the similarity, the shortest distance and the fusion evaluation value. Specific steps are shown in the Table 2.

## 5.3 Performance analysis

The above algorithm results verify the feasibility of the proposed algorithm, and the trust evaluation algorithm can make supply chain enterprise nodes obtain trust tokens which are helpful for trust circulation. Regarding the dimension reduction and focusing capability of trust evaluation data in the above algorithm, this section continues to use the Epinions dataset to test the performance of the proposed recommendation algorithm. Combined with the characteristics of the enterprise number in supply chain, four different samples are tested, each with a different number of nodes: 100, 300, 500, and 1000. To avoid human interference, all nodes and

**Table 2. Trust evaluation algorithm.**

| Algorithm: global trust value |
|---|
| Input: $i$, the evaluated node |
| Output: $v_i$, the trust value of the evaluated node |
| Step 1: use pd.read_csv to read the trust score two-dimensional data table of all nodes of the platform, use fillna() to fill the blank value to obtain the all-node scoring matrix *file* |
| Step 2: if r! = 0, determine whether the node is a direct evaluation node, obtain $S$ and $U$ by traversing, use remove() to eliminate the evaluated node itself in the node set, get the fianal version of $S,U,D$ |
| Step 3: use list(set().intersection()) to get $H$, useloc() to get the initial matrix $R$ |
| Step 4: delete empty columns from the matrix by using dropna(), obtain the $R$ matrix that is used to calculate the similarity |
| Step 5: set $\lambda,\alpha$, step size to train the Funk-SVD model to get the new dense evaluation matrix $R$ |
| Step 6: calculate the similarity $\rho$ of each point of $u$ and $H$, add and multiply to obtain the initial indirect evaluation value $v_{ui}$ |
| Step 7: use file[file! = 0] = 1 and new_sheet.to_csv to get the distance data table between nodes, get the direct evaluation node set of each node list_x, use while u not in list_x to loop calculation to get the shortest distance $Dis$ |
| Step 8: use log() to build an information entropy model, get the evaluation weight $\omega_{ui}$, multiply $\omega_{ui}$ and $v_{si}$ to get the direct evaluation value, in the same way, the indirect trust evaluation value is obtained |
| Step 9: traverse the nodes in $S$ and $U$ in turn and obtain the corresponding direct evaluation value and indirect evaluation value, use np.average to obtain $v_i$ |

**Table 3. Trust matrix density and memory variation.**

| Sample | Density1 | Density2 | memory usage1 | memory usage2 |
|---|---|---|---|---|
| No.1 k = 100 | 6.83% | 100% | 116.1 kb | 664.0 bytes |
| No.2 k = 300 | 4.30% | 100% | 549.7 kb | 2.0 kb |
| No.3 k = 500 | 4.02% | 100% | 959.8 kb | 3.7 kb |
| No.4 k = 1000 | 1.15% | 100% | 6.7 mb | 15.7 kb |

their evaluation information in each sample are randomly extracted by a computer from the Epinions dataset.

To facilitate expression, the number 1 is used to represent the parameter situation before implementing dimensionality reduction and, and the number 2 is used to represent the parameter situation after implementing dimensionality reduction and focusing. RMES represents the root-mean-square error of trust ratings, which is the square root of the sum of the squared differences between the actual values and the predicted values, divided by the number of predictions. The test results are as follows:

Table 3 presents the changes in density of the trust matrix after applying the Funk-SVD algorithm on different sample datasets, as well as the memory usage of the trust data after dimensionality reduction and aggregation of the data. Comparatively, the increase in sample data size not only exacerbates the sparsity of the matrix but also occupies a significant amount of memory space. The dense matrix obtained after applying the Funk-SVD algorithm not only has improved density, but also occupies relatively less memory. This advantage becomes more apparent when dealing with large sample sizes. By comprehensive comparison, we found that the proposed algorithm can fully focus on the trust information in the sparse matrix, and solve the problems of cloud resource waste and platform data management inefficiencym, which is caused by a large number of empty values of the trust matrix occupying the database memory in the supply chain finance platform architecture.

Table 4 lists the trust data error magnitude and algorithm running time under different parameter combinations ("step" and "α") during the training model process. The "step" refers

**Table 4. Trust data error and algorithm running time.**

| Sample | step | α | RMES1 | RMES2 | TIME |
|---|---|---|---|---|---|
| No.1 k = 100 | i = 1000 | 0.001 | 36.2138 | 9.6842 | 1.6060 |
| | i = 3000 | 0.001 | 36.2138 | 9.6831 | 4.7398 |
| | i = 5000 | 0.0001 | 37.5259 | 9.6517 | 8.0106 |
| | i = 7000 | 0.0001 | 37.5259 | 9.6516 | 11.3134 |
| No.2 k = 300 | i = 1000 | 0.001 | 1331.6243 | 11.9799 | 5.4984 |
| | i = 3000 | 0.001 | 2002.2102 | 11.9879 | 15.7592 |
| | i = 2000 | 0.00001 | 2019.2543 | 11.8892 | 10.5776 |
| | i = 1000 | 0.00001 | 2019.2543 | 11.8892 | 5.0747 |
| No.3 k = 500 | i = 1000 | 0.001 | 1444.5097 | 14.6307 | 5.9976 |
| | i = 3000 | 0.001 | 1444.5097 | 14.4145 | 22.6019 |
| | i = 3000 | 0.0001 | 1605.9412 | 12.7540 | 19.5640 |
| | i = 380 | 0.00001 | 1623.6534 | 12.6774 | 2.5180 |
| No.4 k = 1000 | i = 1000 | 0.001 | 1011.5792 | 12.5961 | 22.3691 |
| | i = 3000 | 0.0001 | 1432.7987 | 12.2727 | 66.7439 |
| | i = 500 | 0.0001 | 1432.7987 | 12.2718 | 11.6449 |
| | i = 500 | 0.00001 | 1493.5299 | 12.2519 | 11.0170 |

to the number of iterations and "α" refers to the step size during each optimization step in the process of fitting the initial model to the optimal model. "RMSE1" represents the error after the first iteration of the evaluation matrix. "RMSE2" represents the error value between the new rating matrix after i iterations with the original matrix, after i iterations with the original matrix. "TIME" refers to the program running time needed to converge the loss function to the optimal value through iterations.

By comparing the minimum error and running time of each sample in Table 4, it can be observed that increasing the node information data will result in a longer computation time and a smaller minimum error in calculating the optimal value. However, when the number of samples is the same, increasing the multiple of iterations (i.e., step) has little impact on the minimum error, but it will cause a proportional increase in the running time. In addition, when the number of iterations remains unchanged, doubling the step size has a relatively small impact on the error and running time. Therefore, the proposed algorithm can balance the computation time when dealing with large sample data by reducing the number of iterations and increasing the step size. For example, when i = 380 and α = 0.00001, the error is minimized, and the running time to obtain the optimal evaluation matrix is only 2.5180 seconds in the sample 3. In addition, comparing sample 3 with sample 4, the minimum error that the algorithm can finally achieve when processing large sample data is almost the same as that of small sample data. Therefore, the proposed algorithm has a good calculation speed and good fitting performance.

And by comparing the minimum errors under different numbers of samples, it can be observed that the minimum error does not increase proportionally with the multiplication of sample size, especially when K = 500 and K = 1000, the minimum error remains stable within the range of 11–12. This indirectly verifies that the proposed algorithm maintains a stable level of accuracy when processing sample data within a certain range of quantity.

## 5.4 Model comparison

In order to measure the performance of Funk-SVD trust algorithm fused with information entropy model, it is compared with Funk-SVD algorithm without information entropy model and SVD algorithm fused with information entropy model. The results are shown in Table 5.

Compared with the SVD-L model with long running time and the Funk-SVD model with large information error, Funk-SVD-L can better balance the trust error and running time. In the data test with sample size of 200 and 500, the running time can be greatly reduced with acceptable error by debugging each parameter of Funk-SVD-L. In summary, the performance advantages of the proposed algorithm can be summarized as follows:

1. The algorithm has strong ability of dimension reduction and focus of trust data. The algorithm transforms the sparse trust matrix into dense matrix and saves the storage space occupied by the trust data, which is beneficial to the platform database expansion management.

**Table 5. Trust data error and running time of each model.**

| Sample | Funk-SVD-L | | Funk-SVD | | SVD-L | |
|---|---|---|---|---|---|---|
| | RMES | TIME | RMES | TIME | RMES | TIME |
| k = 200 | 9.8976 | 8.4321 | 13.4532 | 7.3219 | 8.9021 | 23.4521 |
| k = 500 | 12.6774 | 2.5180 | 15.4321 | 3.5491 | 8.4502 | 24.5632 |
| k = 1000 | 12.2519 | 11.0170 | 14.8709 | 11.4354 | 10.0909 | 30.7609 |
| k = 2000 | 14.2340 | 10.3210 | 14.9101 | 11.9032 | 10.5643 | 31.8902 |

2. The algorithm has the characteristics of machine learning, which can continuously reduce the error and improve the accuracy of prediction score after adjusting parameters. The algorithm also has certain flexibility and intelligence, which can supervise the convergence of the loss function in the process of model training, shorten the search time of the optimal solution by adjusting the parameters, and improve the running speed.

3. The algorithm has good ability to deal with large sample data.

In conclusion, the algorithm has advantages in trust evaluation data processing and trust value calculation in supply chain finance. It is suitable for supply chain financial platforms with a large number of nodes and a sparse trust matrix, and it is compatible with the technologies of database access, intelligent modeling and data storage management under the Industrial Internet platform embedded with blockchain, which can help the generation and circulation of supply chain finance trust.

## 6 Suggestions for improving supply chain financial credit evaluation mechanism

The integration of Industrial Internet and blockchain technology provides a favorable environment for the trusted circulation of supply chain finance. The design of a global trust evaluation mechanism provides technical support for effective trust evaluation and consensus. In practical applications, the proposed supply chain finance trust architecture and evaluation mechanism are also influenced by various factors other than technical factors. Therefore, to establish a more complete trust mechanism for industrial Internet of things supply chain finance, further work needs to be done in the following aspects.

### 6.1 Promote supply chain finance platform architecture and deepen the use of financial credit evaluation mechanism

Supply chain finance is rapidly moving towards digitalization. As the supply chain industry continuously expands, the number of upstream and downstream enterprises involved is also increasing. To ensure the efficient operation of supply chain financial platforms, the implementation of a financial credit evaluation mechanism has become particularly important. The credibility of supply chain finance largely relies on the completeness and accuracy of information, and the financial credit evaluation mechanism is key to ensuring this credibility. This mechanism not only strengthens trust between enterprises but also provides them with a more precise reference for financing decisions. Therefore, supply chain financial platforms must enhance their scalability, ensuring seamless integration between networks, devices, data, and enterprises. This integration should encompass all supply chain enterprises, especially those SMEs at the network's edge, guaranteeing their effective access to financial channels.

### 6.2 Establish a security framework of the supply chain finance platform architecture to ensure the effectiveness of the financial credit evaluation mechanism

With the expansion of supply chain import and export trade and the proliferation of malicious software, the security risks of industrial blockchain are constantly increasing. To address these challenges, financial platforms need to integrate OT and IT networks to establish a robust cybersecurity framework. This framework can effectively reduce network latency and data packet loss, ensuring the stability, timeliness, and reliability of data transmission. Simultaneously, by using asymmetric encryption and identity resolution technology, we can enhance

the confidentiality of information, identity verification, and the traceability of assets, ensuring the account and information security of supply chain finance nodes. These security measures will reinforce the effectiveness of the financial credit evaluation mechanism. Furthermore, the supply chain financial platform should also have dynamic monitoring and real-time information updating capabilities, which will assist supply chain enterprises in efficiently overseeing business transaction activities. And then strengthen the technical trust in the supply chain finance platform architecture, ensuring its safe and stable operation.

## 6.3 Establish a supply chain financial ecosystem and strengthen the financial credit evaluation mechanism

The advanced technology of industrial blockchain offers a promising avenue for supply chain finance. While the fusion of industrial blockchain with supply chain finance is still in its infancy and the full integration of information technology and finance remains to be realized, there's immense potential in harnessing the power of data gathered from the various layers of industrial blockchain supply chain finance, such as Infrastructure-as-a-Service, Platform-as-a-Service, and Application-as-a-Service. Strengthening the integration of this data and employing sophisticated big data analytics and modeling techniques can dramatically enhance the financial credit evaluation mechanisms. This would equip supply chain finance stakeholders with deeper insights, smarter risk assessments, advanced transaction monitoring, and more reliable trust feedback services. Such integration not only bolsters the synergy between industrial blockchain and supply chain finance but also supports SMEs in securing financing and advancing smart production initiatives. Ultimately, these measures are instrumental in refining the operations of supply chain enterprises and cultivating a trustworthy ecosystem for supply chain finance.

## 6.4 Improve the supply chain financial policy and standardize the financial credit evaluation mechanisms

Policy is pivotal in steering the direction of supply chain finance, particularly in its application of financial credit evaluation mechanisms. A comprehensive legal framework for supply chain finance can enhance legal trust, thereby stimulating the enthusiasm of investment institutions and SMEs to participate in supply chain finance. Although supply chain finance already has corresponding regulations in place, with the introduction of industrial blockchain and the continuous advancement of supply chain finance, we still need to refine related incentive and penalty policies. The implementation of these policies not only helps create a standardized and healthy supply chain finance environment, but also strengthens the application of financial credit assessment mechanisms to protect the interests of all parties and further consolidate the trust foundation within the supply chain finance.

## 7 Research conclusions and prospects

The incorporation of blockchain into the Industrial Internet can realize data security, information sharing and intelligent financing of supply chain finance. It effectively addresses the trust crisis caused by information asymmetry, data leakage, and defaults in traditional supply chain finance. In particular, it can improve the accessibility of financing for MEs.

Focusing on the issue of supply chain finance trust, this study discusses the suitability and role of trust in supply chain finance enabled by the Industrial Internet embedded with blockchain, and completely considering the various elements of supply chain financial trust, designed a supply chain finance platform architecture. Then a supply chain finance trust

evaluation mechanism based on the Industrial Internet is proposed to promote the trust flow of supply chain finance. However, to additionally leverage the technological advantages of blockchain and the Industrial Internet, it is still necessary to strengthen the deep integration of technology and business, for example, by expanding the coverage of supply chain connectivity, improving information density and quality, and eliminating information islands. At the same time, to improve the efficiency of supply chain finance and enhance trust, government policy support and effective regulations for financing activities are also needed. Of course, this will also be a major focus of our future research.

## Supporting information

**S1 Data.**
(ZIP)

**S1 File.**
(DOCX)

## Author Contributions

**Conceptualization:** Yingmei Jiang.

**Funding acquisition:** Yingmei Jiang.

**Supervision:** Jinyu Wei.

**Validation:** Jinyu Wei.

**Visualization:** Jinyu Wei.

**Writing – original draft:** Yuxin Li.

**Writing – review & editing:** Yingmei Jiang, Yaoxi Liu.

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
