## [Decision Letter · Decision Letter 0]

19 Sep 2023

PONE-D-23-14487Research on trust mechanism of supply chain finance under Industrial Internet embedded with blockchainPLOS ONE

Dear Dr. liu,

Thank you for submitting your manuscript to PLOS ONE. After careful consideration, we feel that it has merit but does not fully meet PLOS ONE’s publication criteria as it currently stands. Therefore, we invite you to submit a revised version of the manuscript that addresses the points raised during the review process.

We look forward to receiving your revised manuscript.

Kind regards,

Sathishkumar Veerappampalayam Easwaramoorthy

Academic Editor

PLOS ONE

Journal Requirements:

"This paper is supported by the 2022 Humanities and Social Science Youth Fund of the Ministry of Education (Grant No. 22YJC630185).The recipient is Xin Yang"             

6.  We note you have included a table to which you do not refer in the text of your manuscript. Please ensure that you refer to Table 2 in your text; if accepted, production will need this reference to link the reader to the Table

Reviewers' comments:

Reviewer's Responses to Questions

**Comments to the Author**

1. Is the manuscript technically sound, and do the data support the conclusions?

Reviewer #1: Yes

Reviewer #2: Partly

2. Has the statistical analysis been performed appropriately and rigorously? 

Reviewer #1: Yes

Reviewer #2: No

3. Have the authors made all data underlying the findings in their manuscript fully available?

Reviewer #1: Yes

Reviewer #2: Yes

4. Is the manuscript presented in an intelligible fashion and written in standard English?

Reviewer #1: Yes

Reviewer #2: Yes

5. Review Comments to the Author

Reviewer #1: （1）In the abstract, the research findings of the paper In the abstract should have quantitative and qualitative conclusions. The research significance needs to be given after the research results.

(2)In the Introduction, the research problem of the paper is not clear. In the introduction, authors should strengthen the description of the research problem. According to the research question, the author should also state the research objectives.

(3) Literature review. I encourage that the authors should provide the research gaps and departure of this study through literature review. But unfortunately, I have not seen the literature review in the paper

（4）Table 3 Trust matrix density and memory variation, What are the parameters based on? Why choose these values? How to overcome the error of the selected data?

(5)What theory back up this study and how has it been validated with your research outcome.

(6) The theoretical contribution and practical significance of the paper are not clear.

(7) I think the conclusions can be supplemented with policy recommendations.

Reviewer #2: Topics related to blockchain applications in supply chain finance continue to grow rapidly. The trust mechanism of supply chain financing under the Industrial Internet embedded in blockchain can provide input and contribution in the application of blockchain technology. However, this article needs to explain several things, namely:

1. Is the empirical method used experimental? or?

2. The data used is also briefly explained. Is it from https://snap.stanford.edu/data/soc-Epinions1.html?

3. It would be better to add graphs, descriptive statistics and hypothesis testing, for example a difference test using the t-test. In conducting performance analysis, this research only displays data and does not explain the meaning of the parameters in the table. For example, table 4, what is meant by RMES? What is meant by Root Mean Square Error (RMSE)?

The paper should add convincing statistical tests in conducting performance analysis.

6. PLOS authors have the option to publish the peer review history of their article (what does this mean?). If published, this will include your full peer review and any attached files.

Reviewer #1: No

Reviewer #2: No

---

## [Author Response · Author response to Decision Letter 0]

23 Jan 2024

Dear Reviewer1,

We would like to express our sincere thanks to the editor and the reviewers for their valuable evaluation and the useful suggestions for our paper. We have revised the paper in the related parts (please refer to the revised manuscript) in accordance with the comments and suggestions of the editor and the reviewers. The following is the summary of how we revised the manuscript in response to the reviewer’s comments.

Point 1: In the abstract, the research findings of the paper In the abstract should have quantitative and qualitative conclusions. The research significance needs to be given after the research results.

Response 1: Thank you for your valuable feedback on the abstract. In light of your suggestions, we have revised the abstract to include both quantitative and qualitative conclusions of our research. The significance of our findings is now clearly stated following the presentation of the results. The following is what we have changed.

Abstract: In traditional supply chain finance, the financing of enterprise mainly relies on the credit segmentation of the core enterprise, resulting in a short trust transmission radius and poor financing ability. The development of Internet technology, while expanding financing channels, has also seen an increasing severity in issues such as information fraud and data breaches, which has further aggravated the trust crisis in supply chain finance. This paper integrates blockchain technology into the industrial internet platform and analyzes the applicability of both in empowering supply chain financial trust. Then a supply chain financial trust framework, which emphasizes information sharing, data security, and trust circulation, is proposed. Furthermore, combined with the theories of Funk-SVD and entropy value, this paper designs a global trust evaluation mechanism that facilitates the trust circulation in supply chain finance and proposes a recommendation algorithm for global trust. With the testing conducted using the Epinions dataset, it is found that the algorithm proposed in this paper has a strong data dimensionality reduction and concentration ability, especially for large sample data, it can obtain more accurate evaluation values with less space occupation, thus enhancing the trust circulation ability of supply chain finance. Finally, the paper puts forward specific policy recommendations for the implementation of the supply chain finance information mechanism, aiming to better improve the financing accessibility of enterprises in supply chain, particularly small and medium-sized enterprises. 

Point 2: In the Introduction, the research problem of the paper is not clear. In the introduction, authors should strengthen the description of the research problem. According to the research question, the author should also state the research objectives.

Response 2: We appreciate your insights regarding the clarity of the research problem in the Introduction. Following your advice, we have expanded the Introduction section to offer a more detailed description of the research problem. We have also explicitly outlined our research objectives to closely align with the research question. The introduction is modified as follows.

For any country, small and medium-sized enterprises (SMEs) are a crucial force that cannot be ignored in its economic development process. In order to support the development of SMEs, countries around the world have also taken a series of measures to improve the accessibility of financing for SMEs. For example, the US government provides financial loans to SMEs at lower interest rates than the market loans through the establishment of fiscal loans, in order to alleviate the burden on enterprises. The German government, as the leading institution, has collaborated with commercial banks, guarantee companies, and policy banks to establish a systematic financing system and regulatory model for SMEs, in order to balance the conflicting interests of various parties in the financing process. The Chinese government has also introduced a series of fiscal measures to guide and establish new financing models for SMEs. As a result, supply chain finance, which aims to support the development of SMEs, has gradually become a topic of public interest and is constantly evolving as a new direction in supply chain theory.

Supply chain finance is a financing model that relies on the credit of the core enterprise and is formed through the collaboration of multiple parties, including the core enterprise, upstream and downstream SMEs, and financial institutions, in order to create a universal, recyclable, and efficient financing model. Compared with traditional financing models, supply chain finance innovatively introduces the transmission of core enterprise credit, which helps to enhance the credit of SMEs. This enables them to obtain more favorable interest rates and shorter financing time, to a certain extent alleviating the financing constraints faced by SMEs. 

However, in the current development process of supply chain finance, the credit risk of SMEs is considered to be one of the most important and severe risks. On the one hand, SMEs often lack core technology and innovation capabilities, resulting in smaller scale, weaker strength, and limited asset quality. This makes it difficult for them to provide sufficient collateral, leading financial institutions to adopt strict financing conditions to control bad debt risks. On the other hand, supply chain companies often hide certain true information or even disseminate incorrect information in their daily transactions due to different purposes. This asymmetric information not only affects the overall operational efficiency of the supply chain but also increases the risk for financial institutions, leading to a crisis of trust in corporate financing. In fact, SMEs still face difficulties in obtaining financing and high financing costs. 

In addition, the rapid development of industrial Internet technology not only provides a massive data source for the comprehensive evaluation of corporate credibility, but also exposes the deficiencies of financing platforms in terms of data processing capabilities. The inability to extract effective information for credit assessment of SMEs from massive data, as well as the lack of accurate and timely communication, significantly hinders the realization of financing for SMEs and ultimately affects their overall development. Hence, in the context of industrial interconnection, accurately transmitting trust data in the supply chain and designing a scientifically rational mechanism for evaluating trust in SMEs is not only crucial for solving the financing problem of these enterprises, but also for promoting the development of the supply chain and the supply chain finance.

Taking the above considerations into account, this article conducts research on the trust issues in supply chain finance by embedding blockchain technology into the Industrial Internet platform. It primarily discusses three questions: What is the architecture of supply chain finance under the integration of blockchain technology and the Industrial Internet platform? What is the trust evaluation mechanism for supply chain finance under this architecture, and how effective is it? How can we promote the implementation of the trust mechanism for supply chain finance by integrating blockchain and the Industrial Internet?

Based on the above issues, this paper comprehensively explores the applicability and role of Industrial Internet technology and blockchain technology in supporting supply chain finance activities, and constructs a supply chain finance platform architecture that integrates blockchain technology and industrial internet. By utilizing the distributed data storage, dynamic transaction monitoring, and massive data modeling and analysis technologies of the Industrial Internet platform and blockchain, a global trust evaluation model that integrates Funk-SVD and information entropy is designed. Flexible and intelligent machine learning recommendation algorithm is used to generate a trust value that includes comprehensive trust information about financing companies. This aims to improve the accuracy of trust assessment for SMEs in supply chain, and alleviate the information asymmetry in supply chain finance. Finally, the proposed algorithm is subjected to performance tests using a publicly available dataset, confirming its superiority in processing large sample data. Based on the analysis of the test results, corresponding countermeasures are proposed.

The main contributions and significance of this study are as follows: Firstly, by integrating blockchain technology with industrial Internet technology, the hierarchical architecture of supply chain financial trust is reconstructed, and the operational logic of the supply chain financial trust mechanism is proposed. Secondly, the recommendation algorithm based on Funk-SVD and information entropy is applied to the evaluation of supply chain financial trust under the industrial Internet, achieving a comprehensive trust evaluation of supply chain trust mechanism. Thirdly, this study proposes strategies and suggestions for improving the supply chain financial trust mechanism under the integration of industrial Internet and blockchain, which can help the supply chain improve the accessibility of financing for SMEs and effectively control the bad debt risk of financial institutions. In general, this research not only enriches the theoretical research of the supply chain and supply chain finance under the integration of the industrial Internet and blockchain, but also provides guidance for the financing practices of SMEs in the supply chain in the Internet era.

Point 3: Literature review. I encourage that the authors should provide the research gaps and departure of this study through literature review. But unfortunately, I have not seen the literature review in the paper.

Response 3: We apologize for the oversight and thank you for your suggestion to identify research gaps through a literature review. We have now included a comprehensive literature review, combing through three areas of research on trust mechanisms of supply chain finance, the application of the industrial Internet to supply chain finance, and the integration of supply chain finance with blockchain technology, outlining the existing research landscape that underlies our research questions, and highlighting the contribution of our work to the existing body of knowledge. See the original for detailed changes.

Point 4: Table 3 Trust matrix density and memory variation, What are the parameters based on? Why choose these values? How to overcome the error of the selected data?

Response 4: We recognize the importance of justifying the parameters chosen for Table 3. In the revised manuscript, we have included a detailed explanation of the basis for selecting these parameters, along with the rationale for choosing the specific values. Additionally, during the performance testing, we conducted a series of experiments by adjusting different combinations of iteration times and precision for varying sample sizes to overcome any potential computational errors caused by the selected data. The original is amended as follows.

The above algorithm results verify the feasibility of the proposed algorithm, and the trust evaluation algorithm can make supply chain enterprise nodes obtain trust tokens which are helpful for trust circulation. Regarding the dimension reduction and focusing capability of trust evaluation data in the above algorithm, this section continues to use the Epinions dataset to test the performance of the proposed recommendation algorithm. Combined with the characteristics of the enterprise number in supply chain, four different samples are tested, each with a different number of nodes: 100, 300, 500, and 1000. To avoid human interference, all nodes and their evaluation information in each sample are randomly extracted by a computer from the Epinions dataset. 

To facilitate expression, the number 1 is used to represent the parameter situation before implementing dimensionality reduction and, and the number 2 is used to represent the parameter situation after implementing dimensionality reduction and focusing. RMES represents the root-mean-square error of trust ratings, which is the square root of the sum of the squared differences between the actual values and the predicted values, divided by the number of predictions.

Point 5: What theory back up this study and how has it been validated with your research outcome.

Response 5: In response to your inquiry about the theoretical underpinning of our study, we have now elaborated on the theoretical framework that guides our research. We also correlated our findings with established theories, thus strengthening the reliability of our research. Improve the document as follows.

4 Design of supply chain finance trust evaluation mechanism based on Industrial Internet platform

The trust evaluation mechanism for supply chain finance in the industrial Internet embedded with blockchain mainly involves three parts: a evaluation principle, a trust consensus rule, and a evaluation algorithm. The trust evaluation principle is the way to obtain trust values of supply chain finance nodes and is the internal logic of the trust evaluation mechanism. The trust consensus rule ensures that all nodes of supply chain finance strictly adhere to the evaluation method. And it is also the premise for the effective circulation of trust values. The design of the evaluation algorithm directly determines the level of the calculated final trust value and embodies the rationality and scientificity of the evaluation mechanism. 

Based on traditional trust evaluation theory, the trust evaluation principle and the consensus rule that are suitable for supply chain finance trust operation logic and blockchain technology features are designed. Furthermore, a global trust evaluation algorithm based on recommendation algorithms in the field of computer science and information entropy theory is proposed. The trust token calculated in this way can serve as a credit credential for financing enterprises to obtain financing from investment institutions, aiding SMEs in the supply chain to enhance their creditworthiness and facilitate trust circulation.

Point 6: The theoretical contribution and practical significance of the paper are not clear.

Response 6: Thank you for pointing out the need to clarify the theoretical contribution and practical significance. We have revised the manuscript to articulate the theoretical advancements our paper offers to the academic field. Furthermore, we have detailed the practical implications of our findings. See the last paragraph of the introduction for specific changes

The main contributions and significance of this study are as follows: Firstly, by integrating blockchain technology with industrial Internet technology, the hierarchical architecture of supply chain financial trust is reconstructed, and the operational logic of the supply chain financial trust mechanism is proposed. Secondly, the recommendation algorithm based on Funk-SVD and information entropy is applied to the evaluation of supply chain financial trust under the industrial Internet, achieving a comprehensive trust evaluation of supply chain trust mechanism. Thirdly, this study proposes strategies and suggestions for improving the supply chain financial trust mechanism under the integration of industrial Internet and blockchain, which can help the supply chain improve the accessibility of financing for SMEs and effectively control the bad debt risk of financial institutions. In general, this research not only enriches the theoretical research of the supply chain and supply chain finance under the integration of the industrial Internet and blockchain, but also provides guidance for the financing practices of SMEs in the supply chain in the Internet era.

Point 7: I think the conclusions can be supplemented with policy recommendations.

Response 7: Your suggestion to include policy recommendations is well-taken. In the revised conclusion, we have added specific policy recommendations that stem from our research findings. The following are additions to the policy recommendations.

6 Suggestions for improving supply chain financial credit evaluation mechanism

The integration of Industrial Internet and blockchain technology provides a favorable environment for the trusted circulation of supply chain finance. The design of a global trust evaluation mechanism provides technical support for effective trust evaluation and consensus. In practical applications, the proposed supply chain finance trust architecture and evaluation mechanism are also influenced by various factors other than technical factors. Therefore, to establish a more complete trust mechanism for industrial Internet of things supply chain finance, further work needs to be done in the following aspects.

6.1 Promote supply chain finance platform architecture and deepen the use of financial credit evaluation mechanism

Supply chain finance is rapidly moving towards digitalization. As the supply chain industry continuously expands, the number of upstream and downstream enterprises involved is also increasing. To ensure the efficient operation of supply chain financial platforms, the implementation of a financial credit evaluation mechanism has become particularly important. The credibility of supply chain finance largely relies on the completeness and accuracy of information, and the financial credit evaluation mechanism is key to ensuring this credibility. This mechanism not only strengthens trust between enterprises but also provides them with a more precise reference for financing decisions. Therefore, supply chain financial platforms must enhance their scalability, ensuring seamless integration between networks, devices, data, and enterprises. This integration should encompass all supply chain enterprises, especially those SMEs at the network's edge, guaranteeing their effective access to financial channels. 

6.2 Establish a security framework of the supply chain finance platform architecture to ensure the effectiveness of the financial credit evaluation mechanism

With the expansion of supply chain import and export trade and the proliferation of malicious software, the security risks of industrial blockchain are constantly increasing. To address these challenges, financial platforms need to integrate OT and IT networks to establish a robust cybersecurity framework. This framework can effectively reduce network latency and data packet loss, ensuring the stability, timeliness, and reliability of data transmission. Simultaneously, by using asymmetric encryption and identity resolution technology, we can enhance the confidentiality of information, identity verification, and the traceability of assets, ensuring the account and information security of supply chain finance nodes. These security measures will reinforce the effectiveness of the financial credit evaluation mechanism. Furthermore, the supply chain financial platform should also have dynamic monitoring and real-time information updating capabilities, which will assist supply chain enterprises in efficiently overseeing business transaction activities. And then strengthen the technical trust in the supply chain finance platform architecture, ensuring its safe and stable operation.

6.3 Establish a supply chain financial ecosystem and strengthen the financial credit evaluation mechanism

The advanced technology of industrial blockchain offers a promising avenue for supply chain finance. While the fusion of industrial blockchain with supply chain finance is still in its infancy and the full integration of information technology and finance remains to be realized, there's immense potential in harnessing the power of data gathered from the various layers of industrial blockchain supply chain finance, such as Infrastructure-as-a-Service, Platform-as-a-Service, and Application-as-a-Service. Strengthening the integration of this data and employing sophisticated big data analytics and modeling techniques can dramatically enhance the financial credit evaluation mechanisms. This would equip supply chain finance stakeholders with deeper insights, smarter risk assessments, advanced transaction monitoring, and more reliable trust feedback services. Such integration not only bolsters the synergy between industrial blockchain and supply chain finance but also supports SMEs in securing financing and advancing smart production initiatives. Ultimately, these measures are instrumental in refining the operations of supply chain enterprises and cultivating a trustworthy ecosystem for supply chain finance.

6.4 Improve the supply chain financial policy and standardize the financial credit evaluation mechanisms

Policy is pivotal in steering the direction of supply chain finance, particularly in its application of financial credit evaluation mechanisms. A comprehensive legal framework for supply chain finance can enhance legal trust, thereby stimulating the enthusiasm of investment institutions and SMEs to participate in supply chain finance. Although supply chain finance already has corresponding regulations in place, with the introduction of industrial blockchain and the continuous advancement of supply chain finance, we still need to refine related incentive and penalty policies. The implementation of these policies not only helps create a standardized and healthy supply chain finance environment, but also strengthens the application of financial credit assessment mechanisms to protect the interests of all parties and further consolidate the trust foundation within the supply chain finance. 

We hope that these revisions adequately address your concerns and improve the quality and clarity of our manuscript. We are grateful for the opportunity to enhance our work with your guidance and look forward to further feedback.

Dear Reviewer2,

We deeply appreciate the time and effort you've dedicated to reviewing our manuscript titled " Research on trust mechanism of supply chain finance under Industrial Internet embedded with blockchain". Your comments provide valuable insights which have enabled us to refine our paper. Here's a summary of our revisions in response to your feedback:

Point 1: 1. Is the empirical method used experimental? or?

Response 1: We are very sorry to cause you any doubt in this respect. In order to explain the method used in this paper and its function, we have re-described it in the introduction, which is detailed as follows.

This paper comprehensively explores the applicability and role of Industrial Internet technology and blockchain technology in supporting supply chain finance activities, and constructs a supply chain finance platform architecture that integrates blockchain technology and industrial internet. By utilizing the distributed data storage, dynamic transaction monitoring, and massive data modeling and analysis technologies of the Industrial Internet platform and blockchain, a global trust evaluation model that integrates Funk-SVD and information entropy is designed. Flexible and intelligent machine learning recommendation algorithm is used to generate a trust value that includes comprehensive trust information about financing companies. This aims to improve the accuracy of trust assessment for SMEs in supply chain, and alleviate the information asymmetry in supply chain finance. Finally, the proposed algorithm is subjected to performance tests using a publicly available dataset, confirming its superiority in processing large sample data. Based on the analysis of the test results, corresponding countermeasures are proposed.

Point 2: The data used is also briefly explained. Is it from https://snap.stanford.edu/data/soc-Epinions1.html?

Response 2: We thank you for your inquiry about the data source. Indeed, the data used in this study was derived from the dataset available at https://snap.stanford.edu/data/soc-Epinions1.html. 

The Epinions dataset is derived from real data of the Epinions online review site, comprising 'ratings_data' and 'trust_data.' Epinions operates akin to an e-commerce site where users rate products, and this data is captured in 'ratings_data.' Other users who have not experienced the product can use these aggregated ratings as a reference for potential purchases. Additionally, Epinions features a 'trust' option, allowing users to specify those they trust, and the ratings from these trusted users are recorded in 'trust_data,' lending greater weight to their reviews. The unique trust network of Epinions has made it a classic dataset for researching trust mechanisms and rating behaviors on e-commerce social platforms, as well as other recommendation algorithms within e-commerce. Similarly, industrial Internet supply chain financial platforms also embody characteristics of supply chain business transaction networks and product-service interactions. Therefore, leveraging the trust and rating data from the Epinions dataset could be instrumental in testing trust evaluation algorithms on these platforms. 

Point 3: It would be better to add graphs, descriptive statistics and hypothesis testing, for example a difference test using the t-test. In conducting performance analysis, this research only displays data and does not explain the meaning of the parameters in the table. For example, table 4, what is meant by RMES? What is meant by Root Mean Square Error (RMSE)?

Response 3: Thank you very much for your suggestion. In this paper, we design a global trust evaluation model which integrates Funk-SVD and information entropy. It aims to improve the accuracy of trust assessment in supply chain finance and alleviate the information asymmetry. Actually, the model in this paper focuses on performance testing and validation of the proposed algorithm, so there is no difference test. Thank you for the suggestion on the research method, which may become one of the directions for our future research on supply chain finance trust assessment. And, when performing performance analysis, this study really only presents the data and does not explain the meaning of the parameters in the table. Now, we have added specific explanations at the relevant section of the paper.

To facilitate expression, the number 1 is used to represent the parameter situation before implementing dimensionality reduction and, and the number 2 is used to represent the parameter situation after implementing dimensionality reduction and focusing. RMES represents the root-mean-square error of trust ratings, which is the square root of the sum of the squared differences between the actual values and the predicted values, divided by the number of predictions.

The "step" refers to the number of iterations and "α" refers to the step size during each optimization step in the process of fitting the initial model to the optimal model. "RMSE1" represents the error after the first iteration of the evaluation matrix. "RMSE2" represents the error value between the new rating matrix after i iterations with the original matrix, after i iterations with the original matrix. "TIME" refers to the program running time needed to converge the loss function to the optimal value through iterations. 

We are grateful for your comprehensive feedback, and we hope that the revisions made will elevate the quality and relevance of our research. We look forward to your further comments and suggestions.

Warm regards,

Yingmei Jiang

Jinhua Polytechnic

---

## [Decision Letter · Decision Letter 1]

5 Feb 2024

Research on trust mechanism of supply chain finance under Industrial Internet embedded with blockchain

PONE-D-23-14487R1

Dear Dr. liu,

We’re pleased to inform you that your manuscript has been judged scientifically suitable for publication and will be formally accepted for publication once it meets all outstanding technical requirements.

Kind regards,

Sathishkumar Veerappampalayam Easwaramoorthy

Academic Editor

PLOS ONE

Additional Editor Comments (optional):

Reviewers' comments:

Reviewer's Responses to Questions

**Comments to the Author**

1. If the authors have adequately addressed your comments raised in a previous round of review and you feel that this manuscript is now acceptable for publication, you may indicate that here to bypass the “Comments to the Author” section, enter your conflict of interest statement in the “Confidential to Editor” section, and submit your "Accept" recommendation.

Reviewer #1: All comments have been addressed

Reviewer #2: All comments have been addressed

2. Is the manuscript technically sound, and do the data support the conclusions?

Reviewer #1: Yes

Reviewer #2: Partly

3. Has the statistical analysis been performed appropriately and rigorously? 

Reviewer #1: Yes

Reviewer #2: N/A

4. Have the authors made all data underlying the findings in their manuscript fully available?

Reviewer #1: Yes

Reviewer #2: Yes

5. Is the manuscript presented in an intelligible fashion and written in standard English?

Reviewer #1: Yes

Reviewer #2: Yes

6. Review Comments to the Author

Reviewer #1: This paper integrates blockchain technology into the industrial internetplatform and analyzes the applicability of both in empowering supply chain financialtrust. The structure of the paper is reasonable and the conclusion is correct.The authors have carefully revised the paper, and I agree that the paper is accepted by the journal.

Reviewer #2: This paper cleverly addresses the pressing challenge of trust in supply chain finance, proposing a novel integration of blockchain technology into the industrial internet to facilitate this. The strength of this research lies in its proposed supply chain financial trust framework that is innovative and timely, and has the potential to be very helpful to small and medium-sized enterprises.

Empirical validation using the Epinions dataset is another advantage, demonstrating the robustness of the recommended algorithm through practical implementation and lending credibility to the scalability of the solution. These data-driven insights significantly add value to this paper by providing a quantifiable foundation for the theoretical framework presented.

However, this paper fails to explore how these concepts translate into a complex global market terrain, where varying regulatory environments and technological inequalities may impact the adoption of blockchain-based models. More concrete policy recommendations supported by case studies or field trials could improve the practical application of this research.

In addition, although this paper provides quantitative analysis with statistical support, this paper does not use inferential statistics such as hypothesis testing which shows that the model used is better than other models.

Overall, this paper is commendable for its theoretical innovation and quantitative analysis, but requires further efforts to bridge the gap between theory and application of rapidly developing blockchain technologies.

7. PLOS authors have the option to publish the peer review history of their article (what does this mean?). If published, this will include your full peer review and any attached files.

Reviewer #1: No

Reviewer #2: No

---

## [Editor Report · Acceptance letter]

16 Mar 2024

PONE-D-23-14487R1 

PLOS ONE

Dear Dr. Liu, 

I'm pleased to inform you that your manuscript has been deemed suitable for publication in PLOS ONE. Congratulations! Your manuscript is now being handed over to our production team.

Kind regards, 

on behalf of

Dr. Sathishkumar Veerappampalayam Easwaramoorthy 

Academic Editor

PLOS ONE